# Microparticle RSV Vaccines Presenting the G Protein CX3C Chemokine Motif in the Context of TLR Signaling Induce Protective Th1 Immune Responses and Prevent Pulmonary Eosinophilia Post-Challenge

**DOI:** 10.3390/vaccines10122078

**Published:** 2022-12-05

**Authors:** Thomas J. Powell, Andrea Jacobs, Jie Tang, Edwin Cardenas, Naveen Palath, Jennifer Daniels, James G. Boyd, Harrison C. Bergeron, Patricia A. Jorquera, Ralph A. Tripp

**Affiliations:** 1Artificial Cell Technologies, 5 Science Park, Suite 13, New Haven, CT 06511, USA; 2Department of Infectious Diseases, College of Veterinary Medicine, University of Georgia, Athens, GA 30602, USA

**Keywords:** RSV, G protein, vaccine, microparticle, CX3C, eosinophils

## Abstract

Layer-by-layer microparticle (LbL-MP) fabrication was used to produce synthetic vaccines presenting a fusion peptide containing RSV G protein CX3C chemokine motif and a CD8 epitope of the RSV matrix protein 2 (GM2) with or without a covalently linked TLR2 agonist (Pam3.GM2). Immunization of BALB/c mice with either GM2 or Pam3.GM2 LbL-MP in the absence of adjuvant elicited G-specific antibody responses and M2-specific CD8+ T-cell responses. Following challenge with RSV, mice immunized with the GM2 LbL-MP vaccine developed a Th2-biased immune response in the lungs with elevated levels of IL-4, IL-5, IL-13, and eotaxin in the bronchoalveolar lavage (BAL) fluid and a pulmonary influx of eosinophils. By comparison, mice immunized with the Pam3.GM2 LbL-MP vaccine had considerably lower to non-detectable levels of the Th2 cytokines and chemokines and very low numbers of eosinophils in the BAL fluid post-RSV challenge. In addition, mice immunized with the Pam3.GM2 LbL-MP also had higher levels of RSV G-specific IgG2a and IgG2b in the post-challenge BAL fluid compared to those immunized with the GM2 LbL-MP vaccine. While both candidates protected mice from infection following challenge, as evidenced by the reduction or elimination of RSV plaques, the inclusion of the TLR2 agonist yielded a more potent antibody response, greater protection, and a clear shift away from Th2/eosinophil responses. Since the failure of formalin-inactivated RSV (FI-RSV) vaccines tested in the 1960s has been hypothesized to be partly due to the ablation of host TLR engagement by the vaccine and inappropriate Th2 responses upon subsequent viral infection, these findings stress the importance of appropriate engagement of the innate immune response during initial exposure to RSV G CX3C.

## 1. Introduction

Respiratory syncytial virus is a leading cause of respiratory infection in infants, the elderly, and immunocompromised patients, resulting in >100,000 hospitalizations in the US each year [1]. RSV infection occurs repeatedly, is seasonal, and does not provide long-lasting protection from reinfection [2,3]. RSV infection can cause severe pulmonary disease characterized by bronchiolitis and pneumonia and may be associated with asthma [4]. While there is a single FDA-approved therapeutic modality to treat RSV infection, the cost is prohibitive [5] and it has limited value unless treatment is initiated in a timely fashion [6,7]. There are no approved RSV vaccines in part due to safety concerns raised by the clinical failure of formalin-inactivated (FI-RSV) vaccines in the 1960s in which vaccinated children were more susceptible to RSV disease than unvaccinated children [8,9,10,11]. Several hypotheses have been proposed for the failure of FI-RSV vaccines, including host production of non-neutralizing antibodies that enhanced infectivity of RSV [12,13] and induction of Th2-type immune responses characterized by pulmonary eosinophil infiltration in the infected vaccinated hosts [14,15]. The underlying mechanism of the inflammatory Th2/eosinophilia response may have been the insufficient engagement of the innate immune system by the FI-RSV vaccine due to formalin-induced disruption of endogenous viral ligands for Toll-like receptors (TLRs) [13,16], such as the TLR4 agonist activity of the viral fusion (F) protein [17].

Various RSV vaccine development efforts are at late stages of clinical development, most of which are focused on the RSV F protein which induces neutralizing antibody responses [18,19]. Various platforms including recombinant protein [20,21], viral vectors [22], and mRNA vectors [23] are being utilized to present RSV F in its prefusion configuration with the goal of inducing neutralizing antibodies that bind to the virus and prevent fusion of the viral membrane with the target host cell membrane. Another vaccine target is the attachment or RSV G protein which contains a conserved CX3C chemokine motif (amino acids 182–186) that binds to the fractalkine receptor, CX3CR1, on host airway epithelial cells [24,25] and can trigger fractalkine-like host cell responses that contribute to inflammatory mechanisms [26]. Antibody-mediated inhibition of CX3C-CX3CR1 interaction during RSV infection has been shown to ameliorate inflammatory responses and reduce disease severity in preclinical models [27,28,29,30]. We have shown that a designed peptide including this region of RSV G, when presented by synthetic nanoparticles produced using layer-by-layer deposition (LbL-NP), can elicit protective immune responses in mice that include RSV-neutralizing antibodies, modified RSV-specific cellular responses, and antibody-mediated inhibition of RSV G fractalkine mimicry [31,32]. The immune responses elicited by RSV G are dependent on engagement of the innate immune system via interaction with TLR2 [33,34]. To test the hypothesis that targeted engagement of TLR2 would improve RSV G vaccine potency and impact the phenotype of immune responses, we produced LbL microparticles (LbL-MP) bearing the RSV G central conserved region epitope and modified with Pam3Cys, a TLR2 ligand. Our results show that inclusion of a TLR2 ligand in the LbL-MP vaccine shifts the post-challenge immune response toward a Th1-type phenotype by reducing Th2 cytokine and chemokine production accompanied by a striking reduction in post-challenge pulmonary eosinophilia. In light of previous studies postulating the involvement of Th2-mediated inflammation and pulmonary eosinophilia in vaccine-enhanced respiratory disease (VERD) associated with the FI-RSV vaccine [14,15,35,36], and the importance of innate immunity in generating protective immunity to respiratory viruses [37,38,39,40], these results provide guidance on design of RSV vaccines that favor protective responses while minimizing the potential for enhanced disease.

## 2. Materials and Methods

Peptide synthesis and microparticle fabrication: Designed peptides (DP) containing the RSV epitopes of interest and a poly-lysine tail, and diagnostic peptides for use in ELISA and ELISPOT, were synthesized as previously described [31,32]. DP sequences are detailed in Table 1. LbL-MP were fabricated as previously described by alternately layering poly-l-glutamic acid (PGA, negative charge) and poly-l-lysine (PLL, positive charge) on 3 µm diameter CaCO_3_ cores to build up a seven-layer base film prior to addition of DP [41]; layering was accomplished manually by centrifugation or in an automated process using tangential flow filtration, a scalable process that is used in the biopharmaceutical industry for the purification of numerous biomolecules [42,43]. All LbL-MP passed the following release specifications before being used in the reported studies: DP content by amino acid analysis (>200 µg/mL bulk, or >133 pg/particle); endotoxin levels by limulus amebocyte assay or recombinant factor C assay (<0.1 EU/µg DP); and size dispersity by dynamic light scattering (<10 µm diameter). There were no physicochemical differences between particles prepared by the manual or automated methods (data not shown). All peptide syntheses and LbL-MP fabrication were performed in the laboratories of Artificial Cell Technologies, Inc., New Haven, CT, USA.

Mice: Specific-pathogen-free, 6–8 week-old female BALB/c mice (Jackson Laboratories, Bar Harbor, ME, USA) were used in all studies. Mice were housed in micro-isolator cages and were fed sterilized water and food ad libitum.

Virus: RSV A2 was propagated in HEp-2 cells (ATCC CCL-23) as described [44]. FI- RSV was prepared as previously described [26] and stored at 4 °C.

Immunization and viral challenge: All in vivo experiments were performed in accordance with the guidelines and approved protocols of the NorthEast Life Sciences Animal Care and Use Committee (IACUC). Mice were immunized by i.m. injection of LbL-MP suspension (based on dose of DP administered), i.m. injection of FI-RSV (10^6^ plaque forming unit [PFU] equivalents), or i.n. infection with RSV (10^6^ PFU). For i.n. infection, mice were anesthetized by i.p. administration of Avertin (2,2,2-tribromoethanol; 180–250 mg/kg; Sigma-Aldrich, St. Louis, MO, USA) and virus was delivered in PBS at a dose of 10^6^ PFU. Unless otherwise noted, mice were immunized on days 0 and 21 (LbL-MP or FI-RSV) or day 0 only (RSV), bled or sacrificed on day 28 for ELISA and ELISPOT analyses, challenged with RSV on day 35, and sacrificed 5–6 days post-challenge for analysis of lung viral burden, antibody/cytokine/chemokine levels, and eosinophilia (Appendix A). Each mouse study was performed at least twice with comparable results between replicate studies. Specificity of immune responses elicited by LbL-MP loaded with either G or M2 DP was confirmed before initiating the studies described below (Appendix A).

Plaque assay: Lungs were aseptically removed from exsanguinated mice at day 5 post-RSV challenge, and individual lung specimens were placed in 2 mL DMEM and homogenized using an Omni TH tissue homogenizer (Omni International, Kennesaw, GA, USA) mechanical dissociator. Following centrifugation at 400× *g* for 10 min at 4 °C, 200 µL/well of homogenate was added to 90% confluent Vero cell (ATCC CCL-81) monolayers in 24-well tissue culture plates. Following adsorption for 2 h at 37 °C, cell monolayers were overlaid with 2% methylcellulose media and incubated at 37 °C for 6 days. After the monolayers were fixed with 60% acetone/40% methanol, the plaques were enumerated by immunostaining with a cocktail of monoclonal antibodies against RSV F protein (clone 131-2A) and RSV G protein (clone 131-2G) followed by a secondary goat anti-mouse IgG antibody conjugated to peroxidase (KPL). Plaques were detected using 200 µL/well of stable DAB (Invitrogen) at room temperature for 30 min and counted.

ELISA: Serum and bronchoalveolar lavage (BAL) antibody responses were detected by ELISA as previously described [31,32] using 96-well high binding plates (Corning Inc., Corning, NY, USA) coated with 20 µg/mL of G_169–198_ peptide or 10^6^ PFU/mL RSV A2. Antibody titers were determined as the last sample dilution that generated an absorbance reading of greater than 0.2. In some cases, isotype-specific detection antibodies were used.

ELISPOT: T cell responses were measured as previously described [31,32]. Briefly, IL-5 or IFNγ matched antibody pairs (R&D Systems, Minneapolis, MN, USA) and 96 well Multiscreen plates (Millipore, Burlington, MA, USA) were used following the manufacturer’s instructions and stimulated with either 5 µg/mL G_183–197_ peptide or 10 µg/mL M2_81–95_ peptide (Table 1). ELISPOTs were counted on an AID EliSpot Reader System. RSV-specific ELISPOT numbers were determined from triplicate wells by subtracting the mean number of ELISPOTs in the wells containing only medium.

BAL fluid and cells: On the indicated days post-infection, a subset of mice from each group was sacrificed and a tracheotomy was performed to collect BAL fluid. The mouse lungs were washed twice with 0.5 mL of cold PBS and the retained BAL (total 1 mL/mouse) was centrifuged at 400× *g* for 5 min at 4 °C. BAL fluid was transferred to new microtubes and stored at 20 °C until multianalyte analysis as described below or ELISA as described above. BAL cells were stained and analyzed by flow cytometry as described below.

Multianalyte cytokine analysis: Cytokine and chemokine content of BAL fluid was assayed by bead-based multianalyte immunoassay (BioLegend LEGENDplex or Luminex xMAP) following the kit manufacturer’s instruction. Briefly, samples were mixed with capture beads and incubated at room temperature for 2 h, washed 3 times, then a biotinylated detection antibody mix was added and incubated for another 2 h. Finally, streptavidin-PE conjugate was added to the wells for another 30 min incubation. Samples were analyzed by flow cytometry and quantitated by comparison to a standard cocktail included with each kit.

Flow cytometry: Cell surface marker expression patterns were used to identify the following cell types: CD4+ and CD8+ T cells, eosinophils (CD45+/SiglecF+/CD11c^low^), and alveolar macrophages (CD45+/SiglecF+/CD11c^high^). Cell suspensions were blocked with FcγIII/II receptor antibody (BD), and subsequently stained with fluorescent antibodies from eBiosciences, BioLegend, or BD Bioscience, i.e., PerCP-conjugated anti-CD45 (30-F11), APC-conjugated anti-CD11c (HL3 or N418), PE-conjugated anti-SiglecF (E50-2440), FITC-conjugated anti-CD3e (145-2C11), APC-conjugated anti-CD4 (RM4-5), and PE-conjugated anti-CD8 (53-6.7), following the antibody supplier’s instructions. Cells were acquired on a Guava easyCyte 8 flow cytometer (Millpore) with data analyzed using Guavasoft software (v 2.7 or 3.1.1). Cells were gated by forward scatter (FSC)/side scatter (SSC) then by CD45+ staining; eosinophils were identified as FSC^high^/SSC^high^/CD45+/SiglecF+/CD11c^low^ (Appendix A).

Statistical analysis: Data were analyzed in GraphPad Prism 9.0 by one-way ANOVA with Bonferroni’s correction. Where noted for small sample size, data were analyzed by Student’s *t*-test.

## 3. Results

Anti-G peptide antibody response following immunization: To confirm the immunological relevance of peptide G_169–198_ as a vaccine target, BALB/c mice were immunized with FI-RSV (i.m. injection of 10^6^ PFU equivalents on days 0 and 21), or with live RSV (i.n. infection with 10^6^ PFU of RSV A2 on day 0). The mice were challenged with RSV on day 35. On days 28 (pre-challenge), 39 (day 4 post-challenge), and 43 (day 8 post-challenge), three mice per group were sacrificed and BAL fluids were collected. After removal of cells, the BAL fluids were analyzed by ELISA against G_169-183_ peptide (Table 1) or whole RSV A2. Figure 1 shows that either immunization with FI-RSV or infection with RSV induced an antibody response recognizing RSV, while only infection with RSV induced an antibody response recognizing the G_169–198_ peptide. This result confirms that the G_169–198_ CX3C epitope selected for incorporation into the LbL-MP in subsequent studies is recognized by antibodies elicited by RSV infection and thus is relevant to the antiviral immune response.

Dose-dependent efficacy elicited by LbL-MP vaccination: A designed peptide (DP) incorporating the G_169–198_ CX3C chemokine motif and the M2_81–95_ epitope was prepared and layered onto microparticles to yield LbL-MP designated GM2 (Table 1). BALB/c mice were immunized on days 0 and 21 with GM2 doses ranging from 10 µg to 1 ng of DP. On day 7 post-boost, mice were bled and G protein-specific serum antibody responses were measured by ELISA. Figure 2A shows a clear dose-dependent antibody response, with titers ranging from 1:3200 to 1:50 in the majority of the mice even at the lowest dose of 1 ng. On day 7 post-boost, three mice/group were sacrificed, and spleen cells were harvested for ELISPOT analysis in which cells were stimulated with a pool of G_169–198_ and RSV M2_81–95_ peptides. Figure 2B shows that both IFNγ and IL-5 responses peak at the 1 µg dose and decrease in a dose-dependent fashion suggesting a peptide-specific effect. The remaining mice were challenged with RSV 14 days post-boost and the lungs were harvested 5 days later to measure viral titers by plaque assay on Vero cells. Figure 2C shows significant (*p* < 0.05) reduction in lung viral burden compared to naïve mice at all doses tested ranging from 59% at the 1 ng dose to complete protection at the 10 µg dose. Although all immunized groups were statistically (*p* < 0.05) different from the naïve group, there was no statistical difference among the 10 µg, 1 µg, or 100 ng groups (*p* > 0.05 in each comparison). Thus, immunization with GM2 LbL-MP elicits dose-dependent humoral and cellular responses and protection from virus challenge.

Impact of TLR engagement on vaccine potency: To determine the impact of TLR engagement on the responses elicited by LbL-MP vaccination, we modified the DP by adding an N-terminal Pam3Cys moiety to it and layered it onto LbL-MP to yield Pam3.GM2 (Table 1). BALB/c mice were immunized on days 0 and 21 with either GM2 or Pam3.GM2 at doses of 1 µg or 30 ng of DP. ELISA, ELISPOT, and post-challenge viral plaque assays were performed as described above. ELISA results in Figure 3A show no difference in antibody titers elicited by either LbL-MP at the higher dose of 1 µg (1:5750 vs. 1:4250, *p* = NS), while at the lower dose the difference was significant (1:1750 vs. 1:140, *p* < 0.05). Figure 3B shows no significant differences in IFNγ spots elicited by the two vaccine candidates, while Pam3.GM2 yielded significantly fewer IL-5 spots than GM2 (*p* < 0.05), demonstrating that appropriate engagement of TLR2 signaling during immunization with LbL-MP primes the host for a lower Th2 response while absence of TLR engagement leads to a more pronounced Th2 response. In addition to generating more potent antibody responses and dominant Th1 responses, immunization with Pam3.GM2 also yielded a trend toward improved efficacy upon challenge with RSV, although the difference among treatment groups did not reach statistical significance (Figure 3C). The difference in T-cell cytokine profiles in Figure 3B prompted us to look more closely at the antibody responses by ELISA, using isotype-specific detection antibodies. The results in Figure 3D show an increase in RSV G-specific IgG2a and IgG2b isotypes in the mice immunized with either dose of Pam3.GM2 compared to lower levels of these isotypes in the GM2-immunized mice. Collectively, these results show that modification of the DP by addition of Pam3Cys yielded both quantitative (increased antibody titers and efficacy) and qualitative (shift in IL-5:IFNγ ELISPOT ratio and in antibody isotype distribution) improvements in the vaccine-induced immune responses.

Reduction in post-challenge Th2 and pulmonary eosinophilia by inclusion of a TLR2 ligand in the vaccine: Our results show that modification of the G_169–198_ CX3C motif by the addition of Pam3Cys shifts the vaccine-induced immune response away from a Th2 phenotype, but do not address the immune phenotype following live virus challenge of vaccinated hosts. This was examined by immunizing BALB/c mice with GM2, Pam3.GM2, FI-RSV, RSV, or not immunized and then challenging the mice with RSV two weeks post-boost. Five days post-challenge, six mice per group were sacrificed and plaque assays were performed to determine lung viral burden; all immunized mice were protected from challenge as was observed previously (data not shown). Six days post-challenge, the remaining mice were sacrificed, and BAL fluid and cells were harvested and assayed to determine antibody, cytokine, and cellular content. ELISA analysis of G-specific antibodies in the BAL fluids detected IgG1 in all immunized groups except the FI-RSV group, while IgG2a was detected in only the Pam3.GM2-immunized mice and at lower levels in the RSV-infected mice (Figure 4A). When the BAL fluids were analyzed for cytokine content by multiplex ELISA (Luminex), an interesting pattern emerged. Levels of Th2-associated cytokines IL-4, IL-5, and IL-13 were consistently higher in the GM2 and FI-RSV groups and lower in the Pam3.GM2 and RSV groups (Figure 4B) while no such correlation was apparent for levels of Th1-associated cytokines IL-2 and TNFα or Th17-associated IL- 17 (Figure 4C). Finally, the BAL cells were stained for markers of eosinophils (EOS, CD45+/SiglecF+/CD11c−) and macrophages (MΦ, CD45+/SiglecF+/CD11c+) and examined by flow cytometry. Each sample was gated on CD45+ cells to focus on leukocytes. Two-color analysis performed on the gated cells showed no change in MΦ profiles among the different treatment groups (data not shown), but a significant increase in the number of EOS in the GM2 and FI-RSV groups compared to the Pam3.GM2 and RSV groups (Figure 4D). These results demonstrate more pronounced Th2/IgG1/EOS+ responses in mice immunized without proper engagement of the innate immune system (GM2 or FI-RSV) compared to a Th1/IgG2a/EOS− phenotype in mice immunized with proper engagement of the innate immune system (Pam3.GM2 or RSV).

Cytokine/chemokine responses and pulmonary eosinophil recruitment post-challenge: To broaden the examination of the post-challenge immune phenotype in the lungs, we expanded the analysis to include additional cytokines and chemokines. BALB/c mice were immunized i.m. on days 0 and 21 with 1 μg of either GM2 or Pam3.GM2. Control groups were immunized with FI-RSV (i.m., days 0 and 21), RSV (i.n., day 0), or not immunized (naïve). All immunized mice and one half of the naïve mice were challenged with RSV on day 35 while the remaining naïve mice were not challenged and were used as baseline controls (naïve not challenged, NNC). Five days following challenge, 6–7 mice/group were sacrificed and lungs were harvested for analysis of antibody levels, viral burden, and cytokine and chemokine levels. Similar to previous results, post-challenge BAL fluids from GM2 and FI-RSV immunized mice had very low levels of G-specific IgG2a while those from Pam3.GM2 and RSV immunized groups had much higher levels of this Th1-associated isotype (Figure 5A). There was no correlation between serum antibody levels and protection from challenge, as all treated groups yielded 83–100% reduction in RSV plaque numbers compared to the naïve challenge groups, although there was a trend toward lower plaque numbers in the Pam3.GM2 group compared to the GM2 group (Figure 5B). Figure 5D–I show the cytokine and chemokine levels in the post-challenge BAL; only those analytes that differed among the treatment groups are shown for clarity. Figure 5D–F show elevated levels of Th2-associated IL-4, IL-5, and IL-13 in the GM2 and FI-RSV groups compared to the naïve, Pam3.GM2 and RSV groups. A different pattern is seen for TARC and MDC (Figure 5G), as both of these Th2-associated markers are elevated in not only the GM2 and FI-RSV group but also in the RSV group compared to the lower levels seen in the Pam3.GM2 group. Opposing expression patterns are seen for MIP-3α which is elevated in the Pam3.GM2 and RSV groups (Figure 5H) and eotaxin which is elevated in the GM2 and FI-RSV groups (Figure 5I). Eight days following RSV challenge, the remaining three mice per group were sacrificed for collection of BAL cells that were analyzed for cellularity using markers for T cells (CD3+/CD4+, CD3+/CD8+), macrophages (CD45+/CD11c+/SiglecF+), and eosinophils (CD45+/CD11c−/SiglecF+). While there were no meaningful differences among the groups in T cell or macrophage numbers (data not shown), there was a marked increase in numbers of eosinophils in the FI-RSV and GM2 groups compared to all other groups (Figure 5C). It is interesting to note the similar patterns of Th2 cytokines, eosinophil-recruiting eotaxin, and eosinophil numbers in the lung fluid, all elevated in the GM2 and FI-RSV groups but not in the Pam3.GM2 and live RSV groups. However, this pattern of correlation is not absolute, as Figure 5G shows a divergence of expression levels of Th2-associated chemokines MDC and TARC between the live RSV group (elevated levels) and the Pam3.GM2 group (lower levels). Thus, a broader view of Th2 patterns reveals that Pam3.GM2 immunization may offer improved outcome even compared to prior infection with RSV.

## 4. Discussion

Slow progress in RSV vaccine development is partially due to VERD associated with FI-RSV vaccination which caused two fatalities and hospitalization of 80% of FI-RSV vaccine recipients naturally infected with RSV in a clinical trial in the 1960s [10,11]. It is important to note that VERD is not exclusive to FI-RSV vaccines as chimeric RSV FG protein vaccines [45], a vaccinia virus RSV G vaccine [46], low doses of RSV F post-fusion vaccines [47], inactivated RSV [13], and others have all resulted in VERD following RSV infection. Numerous studies have since identified putative causes of VERD, including lack of affinity maturation of B cells resulting in low avidity non-neutralizing antibody responses [13] and insufficient engagement of the innate immune response leading to a Th2-dominated response post-infection [36]. It is likely that the excessive or inappropriate inflammatory responses to some vaccines initiate the potential for VERD that is exacerbated by homeostatic imbalances in the responses to RSV challenge [48]. Thus, it is generally agreed that development of a safe and efficacious RSV vaccine is dependent on induction of both neutralizing antibody responses and non-inflammatory cellular responses that protect the host, rather than exacerbate the disease state.

While most RSV vaccine development has focused on the RSV F protein due to high sequence conservation across strains and its dominant role in eliciting virus-neutralizing antibody responses [20,21,22,49,50], the RSV G presents a promising vaccine target as it contains a highly conserved central region containing a CX3C chemokine-like domain that binds to the host CX3CR1 receptor thereby promoting viral infectivity while also triggering Th2-dominated inflammatory responses with eosinophil infiltration [51]. Antibody-mediated inhibition of RSV G binding to CX3CR1 on host airway epithelial cells reduces viral infectivity and mechanisms of VERD including Th2 responses and eosinophil recruitment [27,31,52,53]. Several platforms have been employed to examine the vaccine potential of the RSV G CX3C motif and have shown induction of protective humoral and cellular responses without overt inflammation upon RSV challenge [29,54,55,56]. We have previously reported that presentation of the CX3C chemokine motif of RSV G and a CD8+ epitope from RSV-M2 protein on a synthetic nanoparticle made by layer-by-layer fabrication (LbL-NP) induces antibodies that block the fractalkine chemotactic activity of RSV G and protect the host from infection and RSV replication post-challenge [31,32]. In the continuation of these studies, we prepared LbL-MP and examined more closely the induction of Th1 and Th2 immune responses and how they correlate with protection from infection and ERD.

Having confirmed infection with live RSV elicited antibodies that recognize the CX3C central conserved region of the RSV G protein (Figure 1), we next examined immunogenicity and efficacy of LbL-MP presenting the G CX3C motif and a CD8+ T cell target epitope from M2. This epitope was included due to the preponderance of evidence showing the importance of CD8+ T-cells in blunting RSV replication while limiting undesirable inflammatory responses in the infected lung [57,58,59,60]. Mice immunized with GM2 LbL-MP yielded dose-dependent antibody responses, both Th1 and Th2 cellular responses, and protection from RSV replication following RSV challenge (Figure 2), consistent with results from our previous studies with LbL-NP [31,32]. The dose response revealed a strong association of antibody titer with efficacy, as a vaccine dose of 1 µg was maximally immunogenic and efficacious while a dose of 1 ng elicited detectable immune responses but minimal efficacy. Since it has been shown that coordinate engagement of innate and adaptive immunity elicits optimal vaccine responses [61,62], we sought to increase the potency of our GM2 vaccine by inclusion of an agonist of innate immunity. One method to accomplish this was via the covalent addition of the TLR2 agonist, Pam3Cys, to GM2 prior to layering onto LbL-MP to yield Pam3.GM2 vaccine. Pam3.GM2 elicited significantly higher G-specific antibody titers compared to GM2 (Figure 3A) and complete protection from viral challenge at a 30-fold lower dose (Figure 3C). In addition to the quantitative improvement in immunogenicity and efficacy, we also observed a qualitative improvement in both humoral and cellular immune phenotypes induced by Pam3.GM2 compared to GM2. The cytokine profile primed by GM2 favored Th2 (IL-5 > IFNγ) while the opposite pattern was seen in mice immunized with Pam3.GM2 (IL-5 < IFNγ) (Figure 3B). The antibody responses followed this same trend, as Pam3.GM2 induced higher levels of Th1-associated IgG2a and IgG2b isotypes compared to GM2 (Figure 3D). Thus, the addition of a TLR2 ligand (Pam3Cys) not only increased the potency of the vaccine but also triggered a favorable shift in the phenotype of the immune response toward Th1/IgG2a/IgG2b.

The shift away from Th2-type immune responses following immunization with Pam3.GM2 vaccine suggested that engagement of the innate immune system might prevent inflammatory responses post-infection, as has been reported for respiratory pathogens including RSV and coronavirus [16,37,38,40]. We expanded the investigation of the immune phenotype in immunized mice and focused on post-challenge responses in the lung, since post-infection outcome is better predicted by the local immune response than by the systemic response [63]. In addition to GM2 and Pam3.GM2 vaccines, this study included groups immunized with FI-RSV or RSV to allow a comparison to the canonical VERD-priming (FI-RSV) regimen and host convalescence from natural infection. Immunized mice were challenged with RSV followed by analysis of immune responses in the lung. Multiplex analysis of cytokines in the BAL fluid showed that only GM2 and FI-RSV elicited Th2 cytokines IL-4, IL-5, and IL-13 while Pam3.GM2 and RSV did not (Figure 4B), while all immunogens elicited Th1/Th17 cytokines (Figure 4C). These results show that immunization in the absence of TLR engagement primed the host for an elevated Th2 response post-challenge while engagement of the innate immune system during immunization prevented this phenotype. This pattern was also detected in the BAL RSV-specific antibody responses, as only the Pam3.GM2 and RSV-immunized groups had detectable levels of Th1-associated IgG2a antibodies (Figure 4A). Characterization of cellular infiltrates in the BAL revealed that only the GM2 and FI-RSV groups had elevated numbers of eosinophils (Figure 4D), which correlates with the high levels of Th2-type cytokines such as IL-5 and IL-13 that are known to be involved in recruitment of eosinophils [64,65]. Thus, modification of the GM2 vaccine by covalent linkage of the TLR2 ligand Pam3Cys steered the vaccine-primed immune response away from the inflammatory Th2/eosinophil phenotype and resulted in a phenotype more closely mimicking that elicited by infection with RSV.

In a broader analysis of post-infection lung immune phenotype, it was confirmed that mice immunized with either Pam3.GM2 or RSV developed higher RSV G-specific antibody responses and a broader isotype distribution than did mice immunized with GM2 or FI-RSV (Figure 5A). Although efficacy did not correlate strictly with immunization regimen, the Pam3.GM2-immunized group exhibited higher levels of protection from RSV challenge than the GM2-immunized group (Figure 5B). Post-challenge examination of eosinophil numbers and several Th2-type cytokines and chemokines reinforced the similarity of phenotype elicited by GM2 or FI-RSV vaccination. As previously observed, these two groups had significantly higher numbers of pulmonary eosinophils post-challenge than the Pam3.GM2 group (Figure 5C). Levels of IL-4, IL-5, IL-13, and eotaxin (Figure 5D–F,I) followed the pattern established in the previous study and correlated inversely with antibody levels and directly with pulmonary eosinophilia as previously reported for other respiratory diseases [65,66]. Like Th2 cytokines, eotaxin (CCL11) is associated with eosinophil migration and activation in lung diseases [67,68]. TARC and MDC (Figure 5G) were elevated in the BAL fluid of mice immunized with GM2 or FI-RSV and also in the mice previously infected with RSV, but not in the Pam3.GM2 group. MDC (CCL22) increases expression of Th2-type cytokines and cellular migration [69] and drives eosinophil degranulation [70], while TARC (CCL17) binds to the CCR4 receptor on Th2 cells causing both cytokine secretion and cellular migration [71]. The presence of elevated levels of MDC and TARC in the lungs of RSV convalescent mice post-challenge suggests that even recovery from natural RSV infection may not completely prevent priming for undesirable immune responses upon subsequent infection, while a focused immunization with G CX3C concomitant with engagement of the innate immune system (Pam3.GM2) does effectively prevent post-infection inflammatory responses in the lung while also protecting the host from RSV infection. Finally, expression of MIP-3α followed a pattern that was divergent from the Th2 cytokines and chemokines, as it was elevated in the Pam3.GM2 and RSV groups compared to the GM2 and FI-RSV groups (Figure 5H). Since eotaxin is chemotactic for eosinophils while MIP-3α is predominantly chemotactic for lymphocytes [72] and has been shown to act as a Th1 and CTL adjuvant [73], these contrasting patterns of expression suggest that MIP-3α may be involved in generating a protective Th1/CTL response to RSV infection while eotaxin and other Th2 factors are involved in inflammatory responses leading to VERD.

Previous reports have highlighted the importance of the mode of vaccine presentation of RSV F or G antigens in preventing post-infection VERD in preclinical models. Vaccine engagement of innate immunity via ligands to TLR2 [74] or TLR4 [75,76] or presentation via virus-like particles [77] elicited robust, protective immunity in the absence of post-infection inflammatory responses including pulmonary eosinophilia. Several studies, both preclinical and clinical, have established the association of pulmonary eosinophilia with VERD following infection with respiratory pathogens [78,79], but there remains debate regarding the contribution of eosinophils to the pathogenic mechanisms of RSV VERD. Conflicting reports have suggested that eosinophils contribute to antiviral immune mechanisms [80] but not to VERD [81], or to both inflammation and antiviral mechanisms [68]; similar conflicting roles of eosinophils have been reported for other infectious diseases including filarial infections [82]. Despite this uncertainty concerning the mechanistic contribution of eosinophils to VERD, our results clearly show that proper engagement of the innate immune system upon RSV G vaccination provides essentially complete protection from viral infection and primes the host for a favorable Th1-type response while inhibiting the inflammatory Th2-type response associated with the fractalkine activity of the RSV G protein.

## 5. Conclusions

Effective vaccination against RSV infection will require protection from both viral replication and pulmonary inflammatory immune responses. The RSV G protein is an attractive vaccine target since it plays an important role in the virus infectious cycle and elicits chemokine-like inflammatory responses via interaction with host CX3C fractalkine receptor. Delivery of the RSV G CX3C epitope and a CD8 epitope from RSV M2 protein via microparticles made with layer-by-layer fabrication (GM2 LbL-MP) elicits protective antibody responses but also a Th2-type inflammatory response including eosinophil infiltration into the infected lung. This undesirable Th2/eosinophil response is ablated by immunizing with LbL-MP engineered to actively engage the host innate immune response via TLR2. Thus, Pam3.GM2 LbL-MP provide superior immunogenicity, protection from viral replication, and reduction in inflammatory Th2/eosinophil responses following challenge with RSV.

## Figures and Tables

**Figure 1 vaccines-10-02078-f001:**
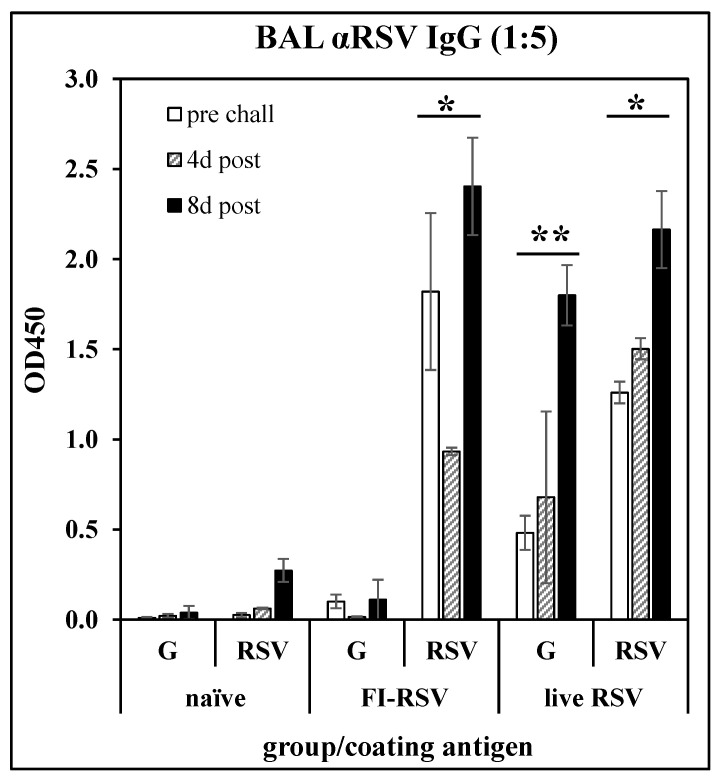
Antibody responses in BAL fluid of mice immunized with FI-RSV or convalescent from RSV infection. BALB/c mice were immunized with FI-RSV on days 0 and 21, with RSV on day 0, or not treated (naïve), then challenged with RSV on day 35. BAL fluid was collected on day 28 (pre-challenge) and 4- or 8-days post-challenge and IgG antibody responses were measured by ELISA against G_169–198_ peptide (G) or whole RSV. Mean ± SEM of 3 mice per group at 1:5 BAL fluid dilutions. * *p* < 0.05 compared to naïve BAL fluid tested against RSV at each time point; ** *p* < 0.05 compared to naïve BAL fluid tested against G_169–198_ peptide at each time point by *t*-test.

**Figure 2 vaccines-10-02078-f002:**
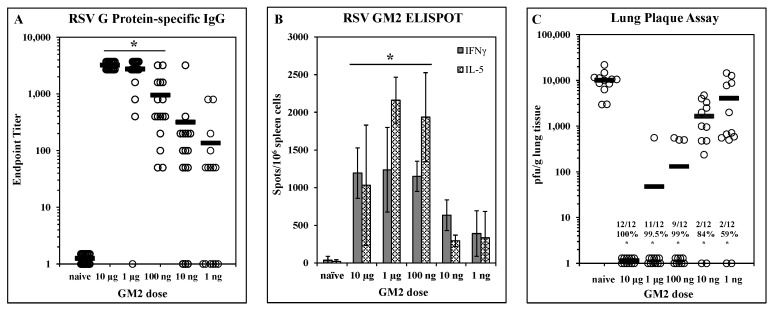
Dose-dependent immunogenicity and efficacy of LbL-MP loaded with RSV GM2 epitopes. BALB/c mice were immunized on days 0 and 21 with the indicated doses of GM2 LbL-MP, challenged with RSV on day 35, and sacrificed for plaque assay on day 40. (**A**) Day 28 post-boost serum antibody titers measured by RSV G protein ELISA. Endpoint titers of individual mice (open circles) and group means (bars). * *p* < 0.05 compared to naïve. (**B**) Day 28 spleen cell ELISPOTs restimulated in vitro with a pool of M2_81–95_ and G_169–198_ peptides. Mean ± SEM spots/10^6^ cells of 3 mice/group. * *p* < 0.05 compared to naïve for both analytes by *t*-test. (**C**) Lung viral titers 5 days post-challenge. Number of plaques per gram of lung tissue for individual mice (circles) and group means (bars). Insets show number of mice with no detectable plaques (*n* = 12 mice per group), average % reduction in plaques per group, and * *p* < 0.05 compared to the naïve group.

**Figure 3 vaccines-10-02078-f003:**
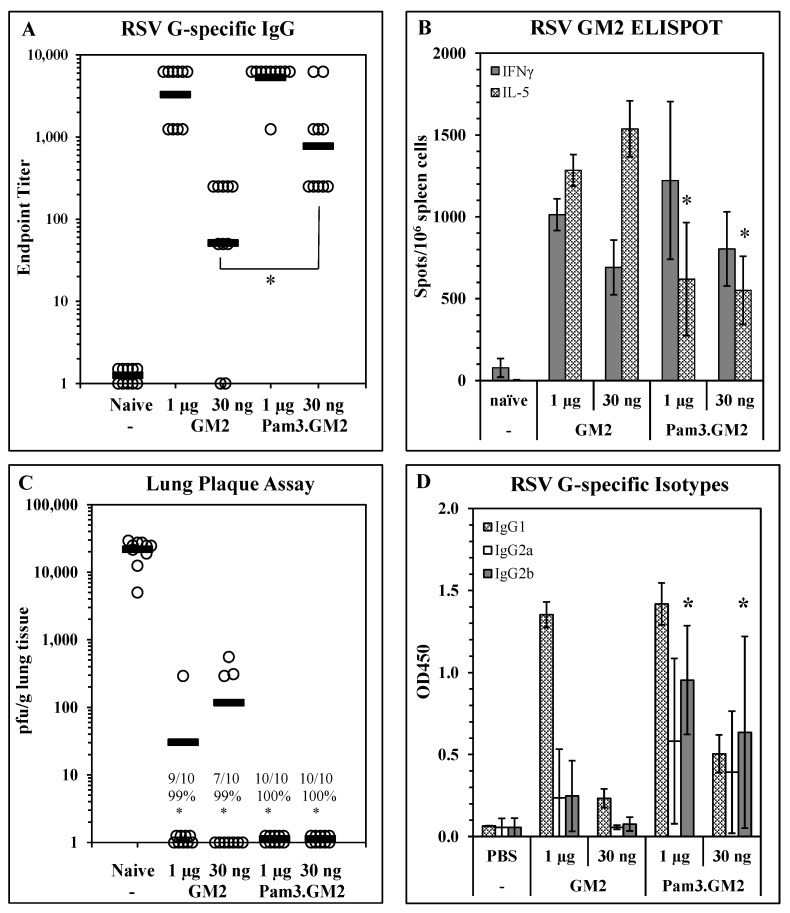
Improvement of immunogenicity and efficacy of LbL-MP by addition of TLR2 ligand Pam3Cys. BALB/c mice were immunized on days 0 and 21 with the indicated doses of GM2 or Pam3.GM2 LbL-MP, challenged with RSV on day 35, and sacrificed for plaque assay on day 40. (**A**) Day 28 post-boost serum antibody titers measured by RSV G protein ELISA. Endpoint titers of individual mice (open circles) and group means (bars); * *p* < 0.05. (**B**) Day 28 spleen cell ELISPOTs restimulated in vitro with a pool of M2_81–95_ and G_169–198_ peptides. Mean ± SEM spots/10^6^ cells of 3 mice/group; * *p* < 0.05 by *t*-test compared to the corresponding dose level of GM2. (**C**) Lung viral titers 5 days post-challenge. Number of plaques per gram of lung tissue for individual mice (circles) and group means (bars). Insets show number of mice with no detectable plaques (*n* = 10 mice per group), average % reduction in plaques per group, and * *p* < 0.05 compared to the naïve group. (**D**) Day 28 serum G-specific antibody isotype levels measured by ELISA. Mean ± SEM of 10 mice per group at 1:50 serum dilution. * *p* < 0.05 compared to the same dose level of GM2.

**Figure 4 vaccines-10-02078-f004:**
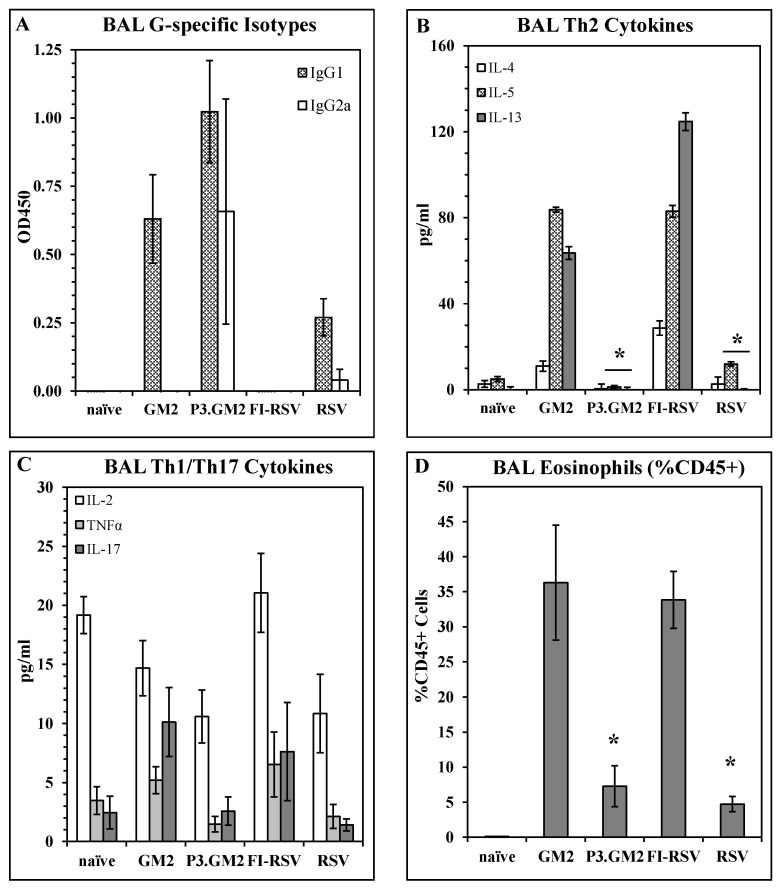
Influence of TLR2 engagement on post-challenge Th2 and eosinophil response. BALB/c mice were immunized on days 0 and 21 as indicated (day 0 only for live RSV), challenged with live RSV on day 35, and sacrificed on day 41 for harvesting of BAL fluid and cells. All analyses were conducted on day 41 (day 6 post-challenge) BAL samples. (**A**) BAL RSV G-specific antibody isotype levels measured by ELISA. Mean ± SEM of 6 mice per group at 1:5 BAL dilution. (**B**) Th2 cytokine content of BAL fluids measured by Luminex. Mean ± SEM of 6 mice per group. * *p* < 0.05 vs. GM2 and FI-RSV groups for IL-5 and IL-13. (**C**) Th1/Th17 cytokine content of BAL fluids measured by Luminex. Mean ± SEM of 6 mice per group. (**D**) Eosinophil counts in BAL. Mean ± SEM of 6 mice per group. * *p* < 0.05 vs. GM2 and FI-RSV groups.

**Figure 5 vaccines-10-02078-f005:**
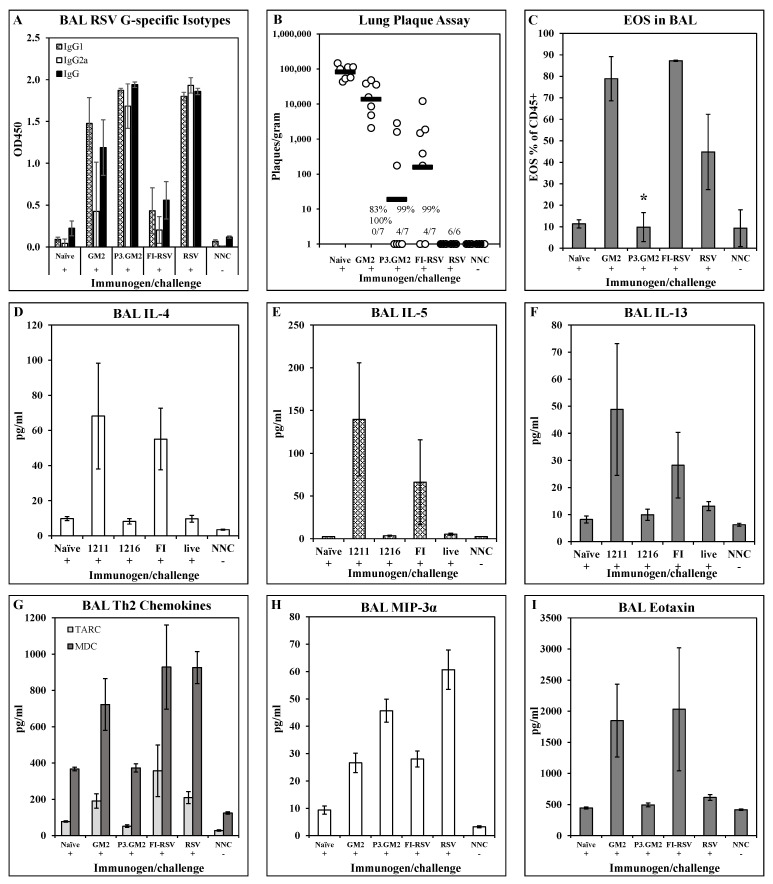
Impact of TLR2 engagement on post-challenge cytokine/chemokine and eosinophilia response in lungs. BALB/c mice were immunized on days 0 and 21 with GM2, Pam3.GM2, or FI-RSV, or infected on day 0 with RSV; control groups were not immunized (naïve and NNC). Mice were challenged with live RSV on day 35 (+) or not challenged (−). NNC = naïve, not challenged. Mice were sacrificed 5 days post-challenge and BAL were harvested for analysis. (**A**) G-specific antibody isotype levels measured by ELISA. Mean ± SEM of 3 mice per group at 1:5 BAL dilution. (**B**) Lung viral burdens. Plaques/gram of lung tissue for individual mice (open circles) and group means (red bars). Insets show number of mice with no detectable plaques (*n* = 6–7 mice per group), average % reduction in plaques per group, and * *p* < 0.05 compared to the naïve group. (**C**) Eosinophil counts in BAL collected on day 8 post-challenge. Mean ± SEM of 6 mice per group. * *p* < 0.05 compared to GM2 and FI-RSV groups. (**D**–**I**) Cytokine and chemokine content of BAL fluids. Mean ± SEM of 6 mice per group.

**Table 1 vaccines-10-02078-t001:** Peptide and particle designs. Pam3CSKKKK = TLR2 ligand. G epitope residues are in bold; M2 epitope residues are underlined; SGS linker residues are italicized; K20 = poly-lysine tail. GM2 and Pam3.GM2 are designed peptides incorporated into the LbL-MP; G_169–198_ and M2_81–95_ peptides are used for ELISA and ELISPOT analyses.

Epitope(s)	Sequence
GM2	** NFVPCSICSNNPTCWAICKRIPNKKPGKKT***SGS*ESYIGSINNITKQSASVA*SGS*K_20_
Pam3.GM2	Pam3CSKKKK**NFVPCSICSNNPTCWAICKRIPNKKPGKKT***SGS*ESYIGSINNITKQSASVA*SGS*K_20_
G_169–198_	** NFVPCSICSNNPTCWAICKRIPNKKPGKKT**
M2_81–95_	ESYIGSINNITKQSA

## Data Availability

All data are reported in the current article.

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
