# Peer review of "Microparticle RSV Vaccines Presenting the G Protein CX3C Chemokine Motif in the Context of TLR Signaling Induce Protective Th1 Immune Responses and Prevent Pulmonary Eosinophilia Post-Challenge"

_vaccines, 2022, doi:10.3390/vaccines10122078_

Round 1
Reviewer 1 Report
The aim of this work was to evaluate the profile of the immune response and the efficacy of a microparticle vaccine conjugated to the G protein CX3C motif, epitopes LT CD8+ and the TLR2 agonist Pam3 in mice challenged with RSV.
The work presents serious deficiencies in the experimental design, in the statistical analysis and in the interpretation of the data. Taken together, these shortcomings prevent the work from being accepted for publication at this time.
Comments:
1. It is unclear why the authors chose the TRL2 agonist. Considering the information provided by the authors, it would be more natural to choose TLR4 to compose the vaccine.
2. Line 173: the authors wrote “This result confirms that the CX3C epitope incorporated into the LbL-MP herein is recognized by antibodies elicited by RSV infection and thus is relevant to the antiviral immune response.”. But LbLMPs were not used in the assays showed in Figure 1.
3. Essential controls are absent from the assays. For example: animals immunized only with LbLMP, LbLMPG and LbLMPM2.
4. Statistical analysis is very poor. The authors do not inform which tests were used. Several results are presented without statistical analysis. In some cases, data interpretation is hampered by the wide standard deviation bars that may have been a result of the low number of animals used.
5. Several times, the authors confuse a mixed pattern of immune response with the polarization of a dominant Th1 response. Example: see line 224, figure 3B.
6. Line 264: the authors said “while IgG2a was detected in only the ACT-1193-immunized mice (Figure 4A).” But, IgG2a was also detected in the FI group.
7. Figure 4C: apparently, there is no difference between all groups, even compared to the naive group.
8. There is no standardization of doses. Authors use 1ug/30ng or 1ug/100ng.
9. Figure 5H: MIP3alpha levels are elevated in all groups, but it is not possible to observe the standard deviation bar and there is no statistical analysis to support the authors' conclusions.
10. Line 330: I didn't understand what the authors meant by this sentence.
11. Line 385: the authors' data seem to be divergent from the literature data as they did not observe any difference in T lymphocyte populations, mainly CD8 TL.
Author Response
- We and others have shown that RSV activates innate immunity through multiple TLRs, including RSV G’s direct effect on TLR2 (Murawski et al. J Virol. 2009;83:1492-500. Alshaghdali et al. Curr Pharm Des. 2021;27:4464-4476; citations added line 72). TLR2 ligand modification of the DP is very straightforward and easily quantifiable, and TLR2 and TLR4 signaling pathways overlap and result in similar biological outcomes.
- We have clarified by changing the original line 173 to read “that the G169-198 CX3C epitope selected for incorporation into the LbL-MP herein in subsequent studies is recognized by antibodies elicited by RSV infection…” in lines 185-188.
- We have tested such controls in numerous studies in the RSV mouse model and other model systems and consistently found that vaccine-induced immune responses were dependent on the inclusion of specific DP on the immunizing LbL-MP. For example, mice immunized with RSV G LbL-MP yielded antibody responses to the included RSV G epitope while mice immunized with RSV M2 LbL-MP did not yield antibody responses to RSV G. The induced antibody response also did not bind to LbL-MP unless that particle had the immunizing antibody target DP. Thus, we did not include those controls in the current studies due to the preponderance of legacy data.
- We apologize for the confusion and now note in the revised manuscript that groups were compared using a one-way analysis of variance (ANOVA) with Bonferroni’s Correction. Data are shown as mean ± SEM. All statistical analyses were performed using GraphPad Prism software by Harrison C. Bergeron in the laboratory of Dr. Tripp who has been added as an author in recognition of his contribution to this important point.
- We agree that what we observed was not a binary switch from exclusively Th1- to exclusively Th2-type, but rather a switch from a mixed Th1 and Th2 response to a largely Th1-type response with an overall reduction in the Th2-type response. We have revised the text in numerous places to reflect this interpretation.
- We apologize for the error in Figure 4A. In fact, IgG2a was not detected in the FI-RSV group. This was incorrectly graphed due to an oversight that has now been corrected to show that IgG2a was detected at robust levels in both ACT-1193 groups, not at all in ACT-1190 or FI-RSV groups, and at lower levels in the RSV group.
- This is a correct interpretation and is relevant to the point stated in the text that Th1/Th17 profiles do not change with or without Pam3Cys.
- All studies used a standard 1 µg dose but did not use the same lower doses. Cross-studies comparisons are facilitated by the 1 µg dose. For the sake of clarity, we have removed the lower dose data from Figure 4 as they do not alter the conclusions of the study.
- We agree and have separated the two analytes into two graphs for easier interpretation (Figures 5H and 5I).
- This sentence has been revised to emphasize the point that in all the other metrics (antibody isotypes, Th2 cytokines, and chemokines), ACT-1216 behaves very similarly to infection with RSV (priming for diminished Th2 responses), but the MDC/TARC data show that some Th2 responses show a divergence between live RSV infection and immunization with Pam3.GM2 LbL-MP. Lines 337-341 in revision.
- While we did not specifically address CD8+ T-cell responses in this report, we have investigated those responses by ELISPOT and in vivo CTL and found that any LbL-MP that contained the M2 CD8 target epitope did indeed elicit CD8+ responses. We did not present the CD8+ responses in the interest of clarity and focus on the balance between Th1 and Th2.
Reviewer 2 Report
Major comments
The present study by Powel group highlighted the beneficial effect of TLR2 agonist adjuvantation and fine tuning the immunogenicity and protective efficacy by using delivery strategy by microparticle. Here are a few major shortcomings listed.
1> The experimental strategy is confusing. Authors should make an effort to provide a schematic schedule describing vaccination regimen, time point for organ harvesting and humoral/cell mediated immune responses. How many mice actually used for the experiment? Are those properly randomized? Did authors make an effort to repeat the experiment and compare the data? How did the authors lock down the number of mice for the experiment? Any power calculations?
2> Did the author actually run any statistics? In Fig 3B, ELISPOT was done with 3 mice and the authors achieved statistical significance in between groups. My interpretation is that the significance was achieved just because of a high responder. It is not biology. Repetition of the experiment needed to make such a claim and gain more confidence.
3> There was a concern regarding data presentation. RSV G protein specific IgG response was measured and presented in Figure 2A/3A as endpoint titer but why is it present as the OD450 value in figure 5A?
4> In figure 1, 2 and 5 data are presented as mean±SD but in figure 4 it was mean±SEM. In figure 3 nothing is mentioned. It is always better to follow only one method, either SD or SEM.
5> Authors claim in the article heading that “Microparticle RSV Vaccines Presenting the G Protein CX3C Chemokine Motif in the Context of TLR Signaling Induce Protective Th1 Immune Responses”. This is too ambitious. I cannot see any Th1 response here. We define Th1 responses if CD4+ T cells produce IFN-g or TNF or IL-2. Authors should run flow cytometry assay to make such a claim.
6> Is ACT 1193 stable? Authors should confirm it by running SEAP assay by using HEK-Blue hTLR2 cells.
7> There should be more characterization of the peptide loaded microparticles. Is it intracellular? If it is-which cell populations are responsible for the update and delivery of the peptides?
8> “Line 153-161”- What is the titration of each antibody? Where is the grating strategy?
Minor comments
There are so many errors. Few of the errors I am listing below.
1> All over the manuscript it is written that the challenge dose is 106 PFU? Is it 106 or 106?
2> Too many punctuation errors. In line 134 it is written as “RSV-M281-95” but in Table 1, it is documented as “M281-95”.
3> “Line 209”- 106 cells or 106?
4> “Line 258”- “Some mice”- It recommended not to use such vague terms in a research article.
Author Response
- We apologize for the confusion and have revised the Results and Figure Legends to clearly define the numbers of mice and time points for each study. We have added a paragraph in Materials and Methods describing the statistical analysis method. Figures 3-5 show the results of three different studies that included the same treatment groups (naïve, GM2, Pam3.GM2 in all three figures; FI-RSV and live RSV added in Figures 4-5), thus allowing cross-study comparison which showed similar patterns in all studies. We conducted additional studies not reported herein that yielded the same pattern of results.
- We note in the revised manuscript that groups were compared using a one-way analysis of variance (ANOVA) with Bonferroni’s Correction. Data are shown as mean ± SEM. All statistical analyses were performed using GraphPad Prism software by Harrison C. Bergeron in the laboratory of Dr. Tripp who has been added as an author in recognition of his contribution to this important point.
- Figure 5 has been modified and no longer includes the serum antibody results in the original Figure 5A since the intent of this Figure was to focus on post-challenge local immune responses in the lungs. For those graphs where antibody isotypes in BAL fluid are shown, the ELISA was carried out at a single BAL fluid dilution of 1:5 and the data are thus reported as OD.
- We have standardized data presentation to mean±SEM for all graphs.
- While IL-2 and IFNγ are produced by both CD4+ and CD8+ cells, they are also markers of Th1-type CD4+ cell activity in contrast to Th2-type markers such as IL-4, IL-5, and IL-13. Figure 4C shows both IL-2 and IFNγ are detected and the pattern of expression does not change with immunogen, in contrast to the Th2-type pattern changes in the same samples as presented in Figure 4B. While we agree that intracellular cytokine staining or cell phenotype depletion studies would definitively address this comment, we strongly believe the data are sufficient to understand that we are seeing a change from a mix of Th1 and Th2 in response to GM2 to a reduction of Th2 in response to Pam3.GM2.
- The LbL-MPs are stable for several months at 4°C when lyophilized and for several weeks at 4°C once reconstituted. The studies reported herein used freshly reconstituted material. While we have not formally investigated which cell populations take up the LbL-MP in vivo, we know from previously published work that LbL-nanoparticles are taken up by dendritic cells (Powell et al. Vaccine 2011;29:558-569, cited).
- Antibodies for flow cytometry were used at the dose recommended by the supplier. A clarification of the gating strategy is now included in the Materials and Methods section Flow Cytometry paragraph.
Minor comments:
- We apologize for the transcriptional error that occurred during the importation of the text into the journal template. The correct dose of 106 pfu is now reported in the text and the Figure legends.
- The nomenclature has now been standardized throughout the text. All LbL-MP are referred to as GM2 or Pam3.GM2 (P3.GM2 in some Figure legends due to space restraints) and all peptides are referred to as G169-198 or M281-95.
- This transcriptional error has been corrected to 106.
- We apologize for this oversight and have corrected the text and the Figure legends to report specific numbers of mice.
Reviewer 3 Report
Comments to authors
In this manuscript, the authors generated microparticle RSV vaccines presenting the G protein CX3C chemokine motif and a CD8 epitope of the RSV matrix protein 2 (GM2) with (Pam3.GM2 LbL-MP) or without TLR2 agonist (GM2 LbL-MP). The authors demonstrated that immunizing mice with the Pam3.GM2 LbL-MP vaccines induced protective Th1 immune responses and prevented pulmonary eosinophilia post-challenge. The findings stress the importance of appropriate engagement of the innate immune response during initial exposure to RSV G CX3C. Pam3.GM2 LbL-MP vaccination provides essentially complete protection from viral infection and primes the host for a favorable Th1 response and inhibits the inflammatory Th2 response.
Listed below are some comments that the authors should address.
1 1. Characterization of the vaccine components: I don’t see the characterization data for the CD8 epitope in Pam3.GM2 LbL-MP. This should be provided.
2 2. Were the GM2, Pam3.GM2, G, and M2 constructs prepared by the authors or by a biotech company? If the latter, the company name should be provided.
3 3. What is ACT? With or without ACT should be the same in all Figures.
4 4. What differences are there between ACT-1190 and ACT-1216, and also between ACT-1193 and ACT-1216? Any explanations as to why different methods for layering were used?
5 5. Please add citations or detailed protocols for the accomplishment of layering for ACT-1190, ACT-1193, ACT-1211, and ACT-1216.
6 6. I’m going to assume the authors meant to state 10^6 pfu (or 10 with superscripted 6). If this is the case, “106” pfu needs to be corrected accordingly throughout the entirety of the manuscript.
7 7. Limitations of the authors’ previous study should be mentioned to emphasize the significance of the present study. If the results of the previous study demonstrated superb protective efficacy, then what is the rationale for extending the authors’ previous works?
8 8. The authors mentioned that “… most focused on the presentation of RSV F” in line 57. Why did the authors choose to work with RSV G protein as the vaccine component in this study? I think that it might be better to briefly explain the advantages and disadvantages of using each of these antigens and if possible, compare them in terms of immunogenicity (Lines 56-65).
9 9. The statistical methods employed in this study should be presented (Line 83) along with the number of biological replicates should be described. Were the data acquired on an individual basis? What statistical analysis was performed and what program did the authors use to calculate the statistical significance between the means? (I’m going to assume ANOVA with post hoc tests or other non-parametric tests, etc).
1 10. If the p-value indicated no significance between groups (Line 220), I don’t think it would be correct to describe the dataset as being particularly higher than another (e.g. “higher antibody titer” in line 219).
1 11. Simply stating “some mice” (line 258) is too generic. Specify the number of mice used for this each experiment.
1 12. Why were different routes of administration used in this study? The authors stated that FI-RSV was intranasally administered, while the vaccines were delivered via IM (lines 295-296)? All inoculums should be administered using an identical route to make accurate comparisons.
Author Response
- Regarding the CD8+ T-cell epitope included in the LbL-MP, we apologize for the confusion but are uncertain about what is being requested. All batches undergo amino acid analysis to confirm peptide content before being released for animal studies. This analysis confirmed that the CD8 epitope was present in all batches tested. In some of the included studies, we confirmed in vivo CTL activity against the CD8+ T-cell epitope elicited by immunization with LbL-MP but did not report it herein since it is not the focus of the report.
- All LbL-MP batches were prepared and quality controlled by the authors at Artificial Cell Technologies (lines 97-100).
- Nomenclature is standardized throughout the text. All LbL-MP are referred to as GM2 or Pam3.GM2 (P3.GM2 in some Figure legends due to space restraints) and all peptides are referred to as G169-198 or M281-95.
- There are no differences in particle composition by amino acid analysis, size by dynamic light scattering, or stability regardless of which method is used. The automated TFF method is simply more efficient and scalable and was developed to support GMP production of clinical-grade material for our most advanced candidate, a malaria vaccine. We routinely use the TFF method now even for research batches.
- Rather than describe the TFF method in detail, since that is not the focus of this report, we note that it is an industry-standard rapid and efficient method for the separation and purification of biomolecules and provide a reference for the general method (Motevalian et al. Biotechnol Progress 2021;37(6):e3204, and Liu et al. ACS Appl Mater Interfaces 2021;13:30326-36, citations added to line 94). The manual method is described in detail as cited in references 31 and 32.
- We apologize for the transcriptional error that occurred during the importation of the text into the journal template. The correct dose of 106 pfu is now reported in the text and the Figure legends.
- Our previous publications did not address the role of innate immunity engagement in vaccine-induced responses. The current study tests the hypothesis that coordinated engagement of the innate and adaptive immune system improves the vaccine-induced response both quantitatively and qualitatively.
- As we mention in the text, the RSV F protein has historically been favored as a target for RSV vaccine development for a number of reasons including greater conservation and a dominant role in eliciting virus-neutralizing antibody responses (Tripp et al. J Virol. 2018;92(3):e01302-17, citation added line 374). Importantly, both RSV F and G proteins can induce neutralizing antibodies and therefore are of interest for the development of vaccines and mAb therapeutics; however, anti-F protein Abs only partially protect from RSV disease that is in part mediated by modified host immune responses contributed by the G protein. For these reasons, we have focused on the G protein , which can induce antibodies that neutralize the virus and protect it from inflammatory disease.
- We apologize for the confusion and now note in the revised manuscript that groups were compared using a one-way analysis of variance (ANOVA) with Bonferroni’s Correction. Data are shown as mean ± SEM. All statistical analyses were performed using GraphPad Prism software by Harrison C. Bergeron in the laboratory of Dr. Tripp who has been added as an author in recognition of his contribution to this important point.
- We agree and have corrected this statement in line 233.
- We have corrected the text and the Figure legends to report specific numbers of mice.
- This appears to be a misreading of the original text, which states, “BALB/c mice were immunized i.m. on days 0 and 21 with 1 μg of either ACT-1211 (GM2) or ACT-1216 (Pam3.GM2). Control groups were immunized with FI-RSV (i.m., days 0 and 21), RSV (i.n., day 0), or not immunized (naïve).” Thus, LbL-MP and FI-RSV were administered i.m. while live RSV was delivered i.n. since that is the appropriate route of infection. Lines 306-309 in the revision.
Round 2
Reviewer 1 Report
The changes and new informantion provided by the authors were of great value. The revision carried out considerably improved the quality of the presentation. I maintain only one point as an impediment to publication.
Although the authors describe consistency of results in control groups, each new preparation of the materials used in the study, each new viral stock, each new batch of animals, each new reagent acquired, must be tested because errors can occur. I consider the presentation of all controls essential. They could be presented as supplementary material.
Author Response
We address the comment regarding the “consistency of results” in two ways. First, we revised lines 94-101 in order to provide the quantitative specifications required before each new lot of LbL-MP is released for in vivo testing. Experimental variation is already addressed by the inclusion of repeat conditions in separate studies. For example, Figure 2 reports the results of immunizing with GM2 LbL-MP, a condition that is repeated in Figure 3 with similar results. Likewise, both Figure 4 and Figure 5 report the results of immunization with GM2 and Pam3.GM2 with similar results between the two Figures (cf. Figure 4B and Figure 5D-F for Th2 cytokines, and Figure 4D and Figure 5C for eosinophil counts). Second, we have added Supplemental Figure 2 with data from a preliminary study that included particles loaded with only G or only M2 DP. Antibodies to G were detected only in mice immunized with G while T cell responses were detected only against the epitope loaded on the immunogen particle for each group, thus demonstrating the specificity of the vaccine-induced immune responses.
Reviewer 2 Report
Major comments
The author's did not address my concerns. So I believe this manuscript should be rejected.
1> No schematic diagram was included for vaccination regimen and experimental strategy.
2> Authors should consult with a statistician. This manuscript contains a serious statistical flaw. With N=3/group, ANOVA is not recommended (Fig. 3B).
3> I believe Flow Cytometry assay with peptide memory recall experiment is needed to address the Th polarization. Also I cannot see any gating strategy for Flow Cytometry. I have a concern over here.
Author Response
1. In addition to the text description in Materials and Methods, “Immunization and viral challenge” paragraph, we have added Supplemental Figure 1 presenting the mouse study design in a schematic format.
2. Figure 3B shows mean±SEM spots/106 cells of 3 mice/per group. As stated in revised lines 122-124, each mouse study was performed at least twice with comparable results between replicate studies. The * in Figure 3B represents p<0.05 by t-test, which is appropriate for small group sizes. Both the “Statistical Analysis” paragraph in Materials and Methods and the legend in Figure 3B have been revised to cite the t-test analysis. Thank you for pointing out this oversight in describing the statistical analysis.
3. The first point is addressed in lines 210-212: “On day 7 post-boost, three mice/group were sacrificed and spleen cells were harvested for ELISPOT analysis in which cells were stimulated with a pool of G169-198 and RSV M281-95 peptides.” We used the ELISPOT method rather than flow cytometry, and the pattern of cytokine responses shows a convincing shift from a balanced Th1/Th2 response to a reduction in the Th2 response whether measured by ELISPOT (Figure 3B) or proteomic analysis of BAL cytokines (Figure 4B and 4C, and Figure 5D-G). The gating strategy for the enumeration of eosinophils is now illustrated in Supplemental Figure 3.
Round 3
Reviewer 1 Report
The authors sent the requested data. I believe the article is ready to be accepted and published.
Author Response
Thank you for your comments.
Reviewer 2 Report
Major comments
The manuscript has serious flaws. Below are my concerns. So I believe this manuscript should be rejected.
1> Manuscript contains statistical flaws. I believe authors should consult with a statistician (I already mentioned it in my second report). Authors ran one-way ANOVA with Bonferroni’s correction for large sample size. For figure 3D or 4A, 5D-I, ANOVA is not recommended though n=6-10 because the values are not normally distributed (as I can see there are high error bars) which is the basic requirement for running ANOVA. The Shapiro-Wilk test is necessary to make such a decision. Are authors aware of such or just randomly ran ANOVA or other statistical method? For each statistical test, there is a logic and I believe authors are not following it.
2> Low sample size is another concern. By using n=3/group, you might get a trend but you should not make a conclusion. There are terms like “blinded experiment”, “random distribution of mice”, “power calculation”. These are necessary to avoid false positives. I am just wondering if authors are aware of such!!
3> As I mentioned in my report 1 and repeating these words here again, by "ELISpot assay and humoral responses"- you cannot draw any concussion about Th1/Th1 biasedness. Please read my comments (Major comment 5) again in report 1.
4> Authors tried Flow Cytometry at least, which is a positive thing. Unfortunately, the Flow Cytometry data is not convincing. I cannot see any live/dead stains. Without it, data could be jeopardized because of the dead cells which are mainly responsible for autofluorescence. The description and data presentation is not up to the mark. Please consult with a Flow Cytometry expert.
Author Response
Reviewer 2 noted that:
- Manuscript contains statistical flaws. I believe authors should consult with a statistician (I already mentioned it in my second report). Authors ran one-way ANOVA with Bonferroni’s correction for large sample size. For figure 3D or 4A, 5D-I, ANOVA is not recommended though n=6-10 because the values are not normally distributed (as I can see there are high error bars) which is the basic requirement for running ANOVA. The Shapiro-Wilk test is necessary to make such a decision. Are authors aware of such or just randomly ran ANOVA or other statistical method? For each statistical test, there is a logic and I believe authors are not following it.
Bonferroni testing is used in a variety of circumstances, and most commonly to correct the experiment-wise error rate when using multiple t-tests or as a post-hoc procedure to correct the family-wise error rate following analysis of variance (ANOVA) (PMID: 24697967). Bonferroni has more power when the number of comparisons is small. We used the Bonferroni post hoc test. The term “post hoc” means “after the event,” thus one uses a post hoc test only after finding a statistically significant result. The requirement here is to determine where the differences come from.
Regarding normal distribution, it seems there is some confusion as the n values per group do not determine if data are normally distributed. For example, Figure 3D contains 10 mice per group, for 5 groups, for a total of 50 mice used in this experiment. Multiple runs, dilutions, and repeats were performed for several hundred data points, indicating the data are normally distributed. Further, high error bars do not automatically preclude normal distribution. Figure 4A does not show significance, confirming that the test is appropriate as it prevented a type I error. Figures 5D-I also include n=7 mice per group, for 6 groups, for multiple technical and independent repeats. Again, hundreds of data points were analyzed, and the data are normally distributed. The authors are aware of basic Gaussian distribution.
- Low sample size is another concern. By using n=3/group, you might get a trend but you should not make a conclusion. There are terms like “blinded experiment”, “random distribution of mice”, “power calculation”. These are necessary to avoid false positives. I am just wondering if authors are aware of such!!
We are aware of potential sample size issues. The terms noted by Reviewer 2 typically reflect larger clinical studies where the main aim of a sample size calculation is to determine the number of participants needed to detect a clinically relevant treatment effect. Pre-study calculation of the required sample size is warranted in the majority of quantitative studies. The factors affecting sample size in this study were study design, method of sampling, and outcome measures. We used the number of observations and independent replicates to estimate the variability that was included in the statistical sample, and we have published this method in >250 publications from our group.
- As I mentioned in my report 1 and repeating these words here again, by "ELISpot assay and humoral responses"- you cannot draw any concussion about Th1/Th1 [sic] Please read my comments (Major comment 5) again in report 1.
Reviewer 2 is incorrect as there are numerous publications that use ELISpot and/or humoral responses to conclude Th1/Th2 biased responses. For example, PubMed indicated >300 manuscripts that address Th1/Th2 biased responses. For historical background on Type 1/Type 2 Immunity in Infectious Diseases please see: https://doi.org/10.1086/317537. These published manuscripts show that one can form conclusions about Th-biased cytokine secretion and IgG subclass switching.
- Authors tried Flow Cytometry at least, which is a positive thing. Unfortunately, the Flow Cytometry data is not convincing. I cannot see any live/dead stains. Without it, data could be jeopardized because of the dead cells which are mainly responsible for autofluorescence. The description and data presentation is not up to the mark. Please consult with a Flow Cytometry expert.
The authors have decades of experience in flow cytometry as evidenced by our publications in PubMed. We are very familiar with the gating criteria, controls, and how to interpret the outcomes. It should be noted that in the third (current) round of review, comment 4 seemed to be combing two or more previous comments from the same reviewer, conflating the eosinophil gating strategy in the supplemental figure with Th1/Th2 staining (“tried Flow Cytometry at least, which is a positive thing”). We had made it clear from the start that we used flow cytometry for eosinophil counts - the critique was that we had not used it for Th1/Th2, and we still haven’t. As previously described, eosinophils were determined by several criteria (FSC/SSC, CD45, CD11c, SiglecF). The same gates were used to compare mice, suggesting autofluorescence which may be occurring should be equal between mice and groups. Finally, eosinophils were gated on a low expressing marker (CD11c), as the reviewer suggests autofluorescence would increase the signal of these events above the ‘low’ gate, excluding them from the final analysis (eosinophils).